TOPICAL REVIEW

# Exercise, healthy ageing, and the potential role of small extracellular vesicles

Luke C. McIlvenna 🔟 and Martin Whitham 🔟

*School of Sport, Exercise and Rehabilitation Sciences, University of Birmingham, Birmingham, UK*

Handling Editors: Ian Forsythe & Paul Greenhaff

The peer review history is available in the Supporting Information section of this article (https://doi.org/10.1113/JP282468#support-information-section).

**Abstract** Extracellular vesicles (EVs) can be released from most cells in the body and act as intercellular messengers transferring information in their cargo to affect cellular function. A growing body of evidence suggests that a subset of EVs, referred to here as 'small extracellular vesicles' (sEVs), can accelerate or slow the processes of ageing and age-related diseases dependent on their

**Luke C. McIlvenna** is currently a postdoctoral researcher at the University of Birmingham in the tissue-cross talk lab, exploring the biological relevance of extracellular vesicles in a variety of physiological contexts. **Martin Whitham** is currently an Associate Professor at the University of Birmingham. He currently leads the tissue-cross talk lab, which investigates the role of small extracellular vesicles in inter-tissue signalling in health and disease. He specialises in the use of mass spectrometry-based proteomics aiming to understand the dynamics of vesicle cargo and surface proteins.

molecular cargo and cellular origin. Continued exploration of the vast complexity of the sEV cargo aims to further characterise these systemic vehicles that may be targeted to ameliorate age-related pathologies. Marked progress in the development of mass spectrometry-based technologies means that it is now possible to characterise a significant proportion of the proteome of sEVs (surface and cargo) via unbiased proteomics. This information is vital for identifying biomarkers and the development of sEV-based therapeutics in the context of ageing. Although exercise and physical activity are prominent features in maintaining health in advancing years, the mechanisms responsible are unclear. A potential mechanism by which plasma sEVs released during exercise could influence ageing and senescence is via the increased delivery of cargo proteins that function as antioxidant enzymes or inhibitors of senescence. These have been observed to increase in sEVs following acute and chronic exercise, as identified via independent interrogation of high coverage, publicly available proteomic datasets. Establishing tropism and exchange of functionally active proteins by these processes represents a promising line of enquiry in implicating sEVs as biologically relevant mediators of the ageing process.

(Received 30 November 2021; accepted after revision 29 March 2022; first published online 7 April 2022)

**Corresponding author** M. Whitham: College of Life and Environmental Sciences, School of Sport, Exercise and Rehabilitation Sciences, University of Birmingham, Edgbaston, Birmingham B15 2TT, UK. Email:m.whitham@bham.ac.uk

**Abstract figure legend** Summary of the role of small extracellular vesicles (sEVs) in the regulation of cellular senescence and the role of exercise derived extracellular vesicles. sEVs are known to play a role in a variety of physiological processes. Recently, they have been identified as potential regulators of paracrine senescence and sEVs from young animals or proliferative cells are capable of reversing/reducing senescence. However, the precise mechanism by which this occurs needs elucidating. As individuals age, the accumulation of senescent cells increases, with this being accompanied by an increase in the senescent associated secretory phenotype and release of sEVs. Exercise is considered to act in a senolytic manner and be capable of preventing or reducing senescence. This subsequently may have implications for age-related pathologies and health span. sEVs may play a role in this process, with antioxidant enzymes and inhibitors of senescence being identified in the cargo of exercise derived sEVs. Created with BioRender.com

## Introduction

Extracellular vesicles (EVs) comprise nano- to microsized particles with a bilipid membrane that are released from all cell types. There are many subtypes of EVs, some of which are formed from either endosomal secretory pathways (exosomes) or are shed from the plasma membrane via outward budding (microparticles). Because contemporary analytical techniques have difficulty in comprehensively separating these subtypes, we collectively refer to them here as small EVs (sEVs). Importantly, sEVs can transfer signalling proteins and other biological cargo such as nucleic acid, metabolites, and lipids between cells and tissues (Stahl & Raposo, 2019; van Niel et al., 2018). They provide a mechanism of protected transport from the harsh extracellular environment and, somewhat challenging the dogma of the endocrinology of secreted proteins, offer a means by which proteins with no signal peptide can be transported outside their cellular origin (Maas et al., 2017). This feature has inspired several lines of enquiry and the examination of the potential role of sEVs in fundamental intercellular communication.

By nature, sEVs are highly dynamic and responsive to physiological stressors, which is reflected in their vast proteome (Iliuk et al., 2020; Whitham et al., 2018). It is becoming more apparent that sEVs play a role in variety of biological processes, with their abundance and cargo being context-dependent. One such process that sEVs may play a role in is cellular senescence, which can be characterised by a state of cell cycle arrest that occurs in response to chronic or acute stress, mediated by internal or external signalling (Lopez-Otin et al., 2013). These stressors include but are not limited to DNA damage, oncogene activation, oxidative stress, and mitochondrial dysfunction (Herranz & Gil, 2018). When these conditions are transient, senescent cells produce a protective response activating signalling to promote tissue repair, immune responses, tumour and suppression, and, subsequently, these senescent cells are cleared by the immune system (Childs et al., 2015). However, when these disruptions in homeostasis are more permanent, senescent cells can accumulate, leading to tissue dysfunction and tumourigenesis (Childs et al., 2015). Although there is currently no single universal marker of cellular senescence, it is recommended that multiple markers are used, encompassing senescence-associated beta-galactosidase (SA $\beta$-gal) activity or lipofuscin (representative of

increased lysosomal activity), cyclin-dependent kinase inhibitors (p16INK4A, p21CIP1 and others), secreted factors, and context-specific factors (Gorgoulis et al., 2019; Sharpless & Sherr, 2015).

Although this process is fundamentally protective, the accumulation of senescent cells is considered to be a driver of ageing and age-related pathologies. Indeed, the life-long or late-life clearance of P16INK4a positive senescent cells in mice can delay or alleviate sarcopenia, cataracts, and loss of subcutaneous adipose tissue (Baker et al., 2011). By contrast, the natural accumulation of P16INK4a positive senescent cells over time results in functional impairments across multiple tissues and organs (Baker et al., 2016). The systemic impact of senescent cells is considered to occur via the release of cytokines, chemokines, and other factors, termed the senescence-associated secretory phenotype (SASP) (Coppe et al., 2008). The SASP can induce senescence in non-senescent cells, often termed paracrine senescence or the bystander effect, via delivery of these factors to neighbouring cells or distant tissues (Acosta et al., 2013).

With the role of senescence as a driver of ageing, it is now becoming a target for the development of several therapeutic strategies to either clear senescent cells (senolytics) or to inhibit the release of SASP factors (senomorphics) (Di Micco et al., 2021). However, many of the current strategies have off-target or side effects that need to be overcome before senolytics or other approaches can be successfully implemented in clinical settings, although these strategies do appear to be promising (Robbins et al., 2021). Exercise may act as a mode of selective senescent cell clearance or in a senomorphic manner. A single bout of resistance exercise can reduce P16INK4a expression in the skeletal muscle of younger individuals, with this effect still observed up to 48 h post-exercise (Yang et al., 2018). As well as local effects of exercise on senescence, more systemic effects have been observed. Following 12 weeks of endurance exercise, older individuals (>60 years of age) display a reduction in the expression of circulating biomarkers of senescence in peripheral blood CD3+ T-cells (Englund et al., 2021). These included p16, p21, tumour necrosis factor-alpha, and cyclic GMP-AMP synthase. In addition to structured exercise training, higher levels of physical activity have been associated with lower levels of P16INK4a in peripheral blood T-cells during ageing (Liu, et al., 2009). A recent systemic review and meta-analysis highlighted that exercise and physical activity can have a senolytic effect via a reduction in P16INK4a positive senescent cells across different immune cell populations (Chen, Yi, et al., 2021). However, the mechanisms by which this effect occurs have not yet been determined. Interestingly, it has been proposed that factors released from active skeletal muscle may modulate these effects by positively regulating immune function (Duggal et al., 2019).

The ageing process can be related to factors, amongst others, that circulate in the blood. This has been shown via heterochronic parabiosis experiments where young and old mice share the same circulation, resulting in an extended lifespan (Ludwig & Elashoff, 1972) and partial restoration of the tissues in the older mice (Villeda et al., 2011). Furthermore, the administration of young plasma into aged mice has been shown to improve age-related cognitive decline (Villeda et al., 2014). More recently, heterochronic parabiosis reduces the expression of SASP factors and senescence in multiple tissues from the skeletal muscle to the brain in old mice. By contrast, the young mice observed the opposite effect with an increase in the expression of senescent markers (Yousefzadeh et al., 2020). However, the mechanisms responsible are unclear, although small EVs have recently been proposed as potential mediators of these effects, regulating the senescent cell burden.

## Small extracellular vesicles as systemic mediators of ageing

**Promotors of senescence.** The increased release of sEVs from senescent cells is not a new concept (Lehmann et al., 2008). However, it is becoming clear that sEVs can modulate senescence, which is dependent on their cargo and cellular origin. Several preclinical studies have determined the functional effects of sEVs, with the initial findings highlighting that sEVs from senescent cells could produce a pro-proliferative effect in cancer cells (Takasugi et al., 2017). This effect was initially identified using conditioned media but, tellingly, the proliferative effect was absent when the conditioned media was depleted of small EVs via ultracentrifugation. Utilising mass spectrometry-based proteomics, a potential small EV cargo protein was identified as being responsible for the effects observed: ephrin type-A receptor 2 (EphA2). This protein is primarily involved in cell proliferation and was the second most abundantly enriched protein in senescent sEVs. It was highlighted that treatment with recombinant EphA2 did not stimulate proliferation in cells, implying that the effect was unique to the delivery of the protein by sEVs. Although genetic ablation of EphA2 eliminated the pro-proliferative effects of the senescent sEVs, it should be noted that 101 proteins were significantly enriched in sEVs from the doxorubicin-induced senescent cells compared to controls, suggesting that other candidate mediators may be involved. It was noted that the increased packaging of EphA2 into sEVs was primarily regulated by an increase in reactive oxygen species (ROS) and could be counteracted by the antioxidant *N*-acetylcysteine. This highlights the role of redox homeostasis in regulating the determinantal effects of the SASP and in sEV cargo sorting.

The SASP can be split into multiple fractions; soluble factors, large EVs and small EVs. Both the soluble

fraction and small EVs from conditioned media of *in vitro* models of oncogene senescence can induce senescence in healthy cells (Borghesan et al., 2019). Treatment with the soluble fraction or the sEVs from senescent cells resulted in a reduction in cell proliferation (bromo-deoxyuridine incorporation), increased expression of cell-cycle inhibitors (p21 and p53), and markers of DNA-damage (p-yH2AX). However, no changes were observed following treatment with large EVs. The induction of senescence was confirmed from size exclusion chromatography (SEC)-isolated sEVs from both oncogene and DNA-damage induced senescence. Further validation was performed by inhibition of sEV release, using inhibitors of neutral sphingomyelinases (Menck et al., 2017), which prevented paracrine senescence induction from conditioned media, thus emphasising the contribution of sEVs to the SASP. There was also a positive correlation between sEV uptake and the activation of paracrine senescence. This may suggest that sEVs from senescent cells have a modified surface proteome (often termed the surfaceome), which creates a preferential affinity/tropism for sEV uptake by non-senescent cells. Again, in an attempt to understand the cargo responsible, mass spectrometry-based proteomics was used in combination with a small interfering RNA screen, identifying interferon-induced transmembrane protein 3 (IFITM3) as a contributor to the induction of senescence via sEVs (Borghesan et al., 2019). Although IFITM3 was enriched in sEVs released from senescent cells, there was no change in the expression in the cell lysate. This highlights the complexity of sEVs in the SASP, moving beyond the idea that sEVs are simply mirroring the cellular state.

**Downstream signalling.** Although the paracrine effect of the sEV SASP is becoming more established, the downstream signalling that mediates this response has not received much attention. To address this significant gap in the field, a small molecule inhibitor screen was performed in the context of oncogene senescence, which identified that the sEV SASP activates transcription factors of the nuclear factor kappa-beta (NF-$\kappa$B) pathway: inhibitor of nuclear factor kappa-B kinase (IKK)$\varepsilon$, IKK$\alpha$ and IKK$\beta$ (Fafián-Labora & O'Loghlen 2021). When the expression of these downstream regulators was suppressed via pharmacological inhibitors or knocked out using a single guide RNA (CRISPR-cas9), it prevented the induction of senescence in human primary foreskin fibroblasts via the sEV SASP. The NF-$\kappa$B pathway is known to be a regulator of cellular senescence and ageing, with small-molecule inhibitor of NF-$\kappa$B activation reducing the accumulation of senescent cells *in vitro* and reducing the expression of the senescence signature in multiple tissues in mouse models of accelerated ageing (Zhang et al., 2021). Given the heterogeneity of the SASP and the development of

cellular senescence, it remains to be determined whether the paracrine effect of the sEV SASP from different models of senescence exerts its effects via independent or common downstream pathways.

**Reversing the senescent phenotype.** By contrast to the transfer of adverse effects of the sEV SASP, it has been observed that sEVs derived from human induced pluripotent stem cells (iPSCs) can reverse the ageing phenotype in senescent cells (Liu et al., 2019). In this instance, sEVs were isolated from conditioned media using ultrafiltration followed by SEC and applied to DNA damage-induced senescent mesenchymal stem cells at a dose of 10 000 EVs per cell. This resulted in a reduction in the levels of ROS, as well as a reduction in SA $\beta$-gal activity and the expression of cyclin-dependent kinase inhibitor proteins (p21 and p53). Mass spectrometry-based proteomics was used to identify a potential mechanism and the cargo responsible, with this anti-ageing effect attributed to peroxiredoxins, comprising a group of antioxidant enzymes enriched in the sEVs from young iPSCs and decreased in senescent cells. The expression of the peroxiredoxins was significantly reduced in the senescent cells, which was accompanied by increase in oxidative stress. Interestingly, peroxiredoxins have been shown to act as inhibitors of cellular senescence (Han et al., 2005; Park et al., 2017) and have been associated with lifespan and stress resistance (Olahova et al., 2008). A deficiency in peroxiredoxins in mice has been shown to worsen skeletal muscle insulin resistance and decreases in muscle strength (Cha et al., 2019; Kim et al., 2018). Further work has highlighted the regenerative potential of stem-cell-derived sEVs (human or young mice) to reduce senescence and increase lifespan in naturally and genetically aged mice (Dorronosoro et al., 2021). A crucial observation was that this effect occurred with just two intraperitoneal injections ($10^9$/EVs) and was comparable to known suppressors of senescence, dasatinib + quercetin and Bcl-2 inhibitor navitoclax, which require more frequent dosing.

Another regulatory pathway implicated in the severity of the pro-inflammatory SASP is NAD+ metabolism. When the rate-limiting enzyme nicotinamide phosphoribosyltransferase (NAMPT) is downregulated or inhibited, this is sufficient to induce senescence (Nacarelli et al., 2019). In addition, NAD+ and NAMPT levels are known to decrease with ageing (Covarrubias et al., 2021). Of relevance, extracellular NAMPT has been found to be contained in sEVs in its active form and capable of increasing NAD+ biosynthesis across multiple tissues (Yoshida et al., 2019). The presence of eNAMPT in sEVs was confirmed in plasma from humans and mice isolated via precipitation, ultracentrifugation, and a density gradient. Given that there is no gold standard method to isolate sEVs, the ability to demonstrate the presence or

a change of expression of proteins via multiple methods of isolation adds strength to the findings. To further demonstrate the role of sEVs in ageing, Yoshida et al. (2019) isolated sEVs from 500 μL of plasma obtained from young mice, which were subsequently delivered to aged mice via intraperitoneal injection. This resulted in improvements in physical activity levels and lifespan (Yoshida et al., 2019). To confirm the role of eNAMPT in this effect, conditioned media from healthy adipocytes and eNAMPT knockdown adipocytes was injected into old mice, with only sEVs from healthy adipocytes producing the anti-ageing effects. Utilising the naturally occurring antioxidant potential of sEVs from younger donors or engineering sEVs to be loaded with anti-oxidant enzymes may be a helpful strategy for mitigating excess oxidative stress and subsequent cellular damage in older individuals. Recently, engineered sEVs have been harnessed to mitigate systemic inflammation in mice (a similar physiological state observed in advance ageing) by enriching the surface of exogenous sEVs with tumour necrosis factor receptor 1 and interleukin-6 signal transducer to act as decoys (Gupta et al., 2021). This produced an anti-inflammatory effect by allowing sEVs to capture these pro-inflammatory cytokines and for them to be subsequently removed from the circulation.

Further evidence that sEVs exert anti-ageing effects via modulating redox homeostasis has been provided by Fafián-Labora et al. (2020), who identified that small EVs from young cells can reduce levels of reactive oxygen species, lipid peroxidation, and DNA damage both *in vitro* and *in vivo*. This was linked to an increase in glutathione and glutathione *S*-transferase (GST) activity, which helps protect against ROS associated tissue damage. Although both the soluble fraction and sEVs can induce senescence, only the sEVs are able to reverse the senescent phenotype. Crucially, the effects of two-weekly intraperitoneal injections of 20 μg of sEVs from young cells for 3 weeks had systemic effects, with SA B-gal expression being reduced in the liver, brown adipose tissue, lung, and kidney. It was noted that sEVs from young cells were enriched in glutathione-related proteins, in particular GSTM2 and GSTA5 as determined via mass spectrometry-based proteomics (Fafián-Labora et al., 2020). In addition, they were able to restore the antioxidant activity in sEVs obtained from old cells by transient transfection of recombinant GSTM2. This presents the possibility of sEV-based therapeutics utilising sEVs from young donors or engineered sEVs from cells with personalised cargo, in particular antioxidant enzymes. Crucially, the exogenous delivery of sEVs derived from HEK293 cells and delivered to other cell types or mice does not appear to cause toxicity or immune responses (Saleh et al., 2019; Zhu et al., 2017). However, it has been noted that as sEVs have the potential to carry pathogens and can promote oncogenesis; therefore, the safety of each EV formulation from different sources (e.g. cell types or human donors) should undergo rigorous safety testing (Herrmann et al., 2021).

This evidence summarised in Fig. 1 highlights the role of sEVs in senescence and the therapeutic potential of sEVs in age-related pathologies by reducing the accumulation of senescent cells and/or reversing the aged phenotype. These senostatic and senolytic effects appear to be partly regulated by the antioxidant potential of sEVs from young, healthy donors, with the metabolic activity of these enzymes being preserved in sEVs, allowing for systemic or targeted delivery. A limitation of the current findings is that they have focused on sEVs from a single cell type that does not account for the complexity of the *in vivo* environment where sEVs are conceivably derived from multiple tissues and cell types.

**Translational challenges.** There are a few challenges that need to be overcome before sEVs can become a viable treatment for ageing: further characterisation of the cargo responsible, identification of the source of the sEVs, purity of sEVs from current isolation methods, routes/mode of administration, and chronic viability (Rodriguez-Navarro et al., 2020). For example, of the studies highlighted in this section, the doses of sEVs used varied between a fixed number, specific protein concentrations or sEVs isolated from fixed volume of plasma. Another critical aspect of sEV research is the respective isolation method used. There is currently a lack of standardised methods for isolating sEVs, with an ever-growing number of methods available. These currently include but are not limited to ultracentrifugation, SEC, density gradient, immunoprecipitation, filter concentration, and precipitation. All isolation methods represent a trade-off between recovery and purity, with higher yielding approaches limited by high levels of contaminant proteins and more specific methods offering limited material for analysis (Cocozza et al., 2020). It is also feasible that some methods alter the EV composition/function, which can have a large bearing on interpretation. Encouragingly, novel approaches to isolate purified sEVs are continuing to be developed for overcoming some of these challenges. For example, chemical affinity isolation anchors specific components of the bilipid membrane rather than traditional targeting proteins such as core tetraspanins (CD63, CD81, CD9), which are often expressed purely in subpopulations rather than being homogeneously expressed in all sEVs, allowing for high yields and purity (Iliuk et al., 2020; Kugeratski et al., 2021). In particular, this approach has been shown improve protein and phospho-protein detection via proteomics compared to ultracentrifugation (Iliuk et al., 2020). Furthermore, instrumentation is being developed for high throughput, efficient isolation of sEVs from biological fluids, which stands to accelerate clarification on the dynamic molecular cargo

of sEVs in the context of health and disease (Chen, Zhu, et al., 2021).

In the context of the SASP, it has been observed that the use of ultracentrifugation alone causes SASP components to be co-isolated with sEVs (Wallis et al., 2021). However, when ultracentrifugation was followed by SEC, this effect could be minimised, allowing for separation of the sEV SASP and co-isolated SASP. Although much focus and debate has occurred in relation to the isolation method, a comparison of the proteome from sEVs isolated via ultracentrifugation, density gradient, and SEC revealed around 70% overlap in the proteins identified (Kugeratski et al., 2021). In addition, it was noted that the cellular origin of sEVs could be used to distinguish the sEV proteome. This may indicate that differences observed in the sEV cargo or functional effects depend more on the cell types used and the method of senescence induction rather than the isolation method selected. The current lack of standardised sEV research methods reinforces the need

for transparent and detailed reporting of the methods following the MISEV guidelines (Théry et al., 2018) and utilising EV-TRACK to asses quality of methods applied (http://evtrack.org) (Van Deun et al., 2017).

## Characterising senescent small extracellular vesicles via mass spectrometry-based proteomics

The biological role of sEVs in the context of ageing and senescence is receiving more attention; further characterisation of the cargo responsible will prove useful for developing therapeutics and identifying markers of ageing/senescence. As already highlighted from the *in vitro* and *in vivo* studies conducted to date, many have utilised mass spectrometry-based proteomics intending to identifying regulators of the functional responses observed. Crucially, these findings require further translation. In clinical settings, plasma samples are obtained

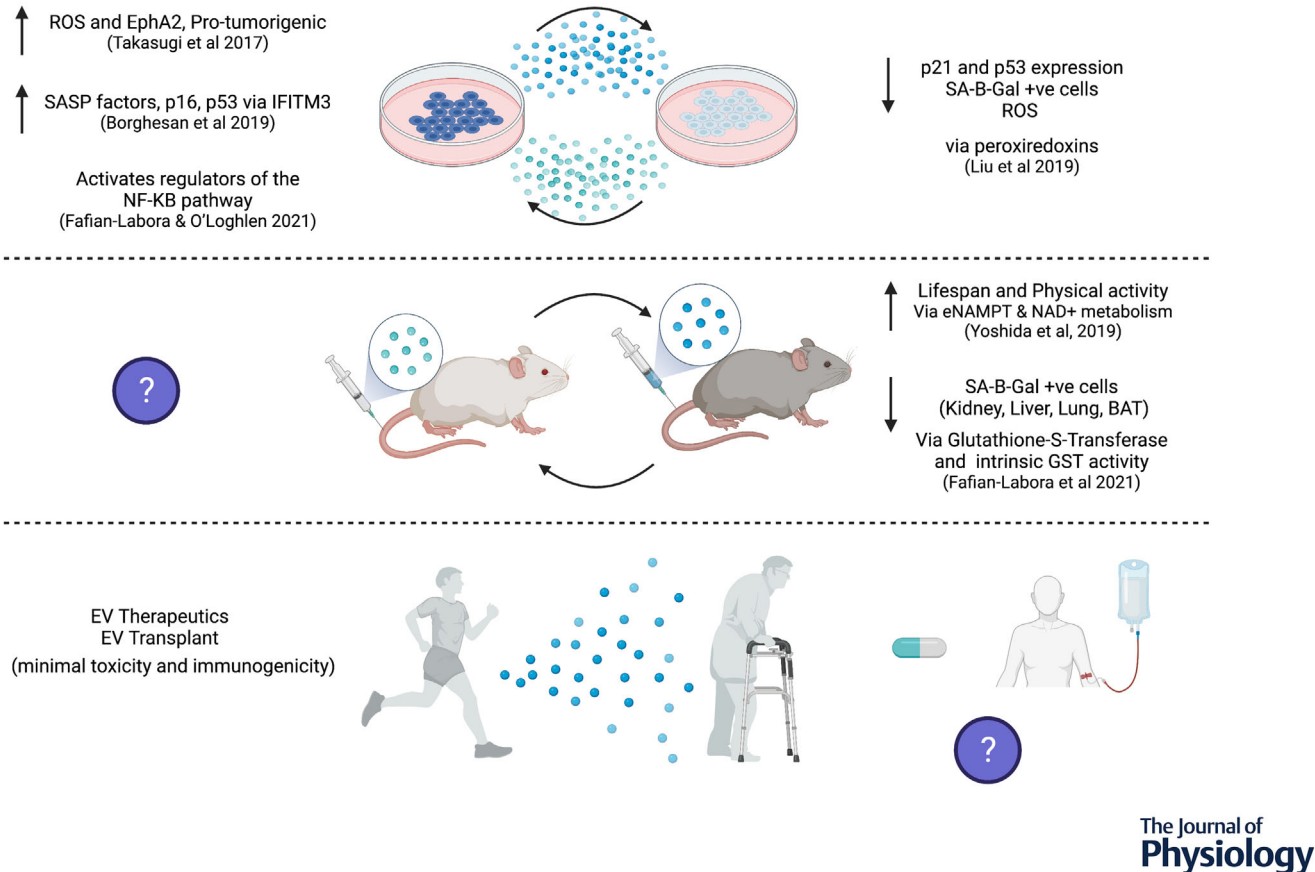

**Figure 1. Role of sEVs in cellular senescence**
A summary of current findings from both *in vitro* and *in vivo* studies. Several studies have demonstrated the effects of transferring sEVs between healthy and senescent cells *in vitro*. The transfer of young sEVs to old mice can partially reverse the senescent phenotype and extend both life and healthspan. The transfer of sEVs from old mice to young mice has still to be studied; however, based upon current evidence, it could be hypothesised that this would lead to an increase in senescent cells and tissue damage. Preclinical work opens the possibilities for sEV therapeutics where the benefits of youth and or exercise could be transferred via sEV transplants or engineered sEVs. Created with BioRender.com.

regularly, with the progress of mass spectrometry-based proteomics (Geyer et al., 2019) providing a powerful tool for exploring the role of the secretome and sEVs in humans as we age. The plasma proteomic signature has been shown to be capable of predicting age-related outcomes and the rate of ageing (Lehallier et al., 2019; Tanaka et al., 2020). Interestingly, plasma sEV concentration and total sEV protein content appear relatively stable across the human lifespan in both males and females (20–85 years old) (Grenier-Pleau et al., 2020). However, the proteomic profile of sEV cargo changes with an increase in proteins related to immune responses and cell adhesion (Grenier-Pleau et al., 2020), which are considered important regulators of ageing.

It has been highlighted that the proteins involved in classical secretion (i.e. those with a known signal peptide) do not account for the whole of the circulating proteome (Whitham & Febbraio, 2019). Indeed, of the 5000 proteins identified in sEVs, only 16% had a signal peptide (Whitham & Febbraio, 2019; Whitham et al., 2018). Given many of the current biomarkers of health and disease have been derived from classical secreted proteins, this leaves many biological relevant markers unaccounted for. When the proteins that make up the traditional SASP (Acosta et al., 2013) were compared with those derived from senescent sEVs, there was little cross-over (Borghesan et al., 2019). Both the soluble factors and sEVs can regulate senescence, demonstrating the need to characterise both to unpack the dynamic and complex nature of the SASP. A more extensive review on use of proteomics in the context of the SASP is provided in Basisty et al. (2020).

Efforts have been made to establish common markers of senescence in the SASP by using several different models of *in vitro* senescence and exploring the associated proteomes: soluble fraction and sEVs. To identify common markers involved in the SASP, Özcan et al. (2016) used five separate senescent inducers (i.e. oxidative stress, doxorubicin treatment, high and low doses of irradiation, and replicative exhaustion) to create the senescent phenotype in the bone marrow and adipose mesenchymal stromal cells. Using proteomics, they characterised the SASP in the conditioned media for each inducer, identifying three key pathways that were present across all phenotypes and associated with features of senescence (MMP2-TIMP2: ECM remodelling; SERPINE1-IGFBP3: paracrine senescence; and PRDX6-PARK7-ERP46-MVP-CTSD: apoptosis resistance). As previously highlighted, this only tells part of the story and conceivably overlooks the significant role of sEVs. More recently, the contribution of both the soluble and sEV proteins to the SASP has been characterised in response to irradiation and oncogene activation using data-independent acquisition proteomics (Basisty et al., 2020). Of the thousands of proteins found

in the senescent sEVs, only nine differentially expressed proteins (ANXA1, ANXA2, ENO3, AHNAK, SLC1A5, ITGA1, COL6A1, COL6A2, and COL6A3) were common to both inducers and the changes in sEV proteins were distinct from the soluble proteins. From this, an excellent resource in the SASP Atlas (www.SASPAtlas.com) was established, which is an online database created from publicly available proteomic datasets that have data from different *in vitro* senescence inducers and cell types for the soluble and sEV proteins (Basisty et al., 2020). Crucially, the identified biomarkers and regulators of senescence identified require *in vivo* validation. Another concept that has yet to be explored is whether the plasma sEV proteome could detect phenotypic changes prior to senescence occurring.

To improve the quality and reproducibility of sEV proteomics data, lessons can be drawn from the already well-established plasma proteomics community, such as detailed reporting of sample collection and preparation, using quality assurance marker panels to determine contamination, and, where possible, avoidance of pooling samples for analysis (Deutsch et al., 2021). Although sample preparation and isolation methods remain a confounder, there are several advantages to utilising in data-dependent acquisition mass spectrometry-based proteomics in the context of sEVs. It allows for an unbiased and hypothesis free approach, enabling researchers to answer specific questions at the same time as accelerating discovery by identifying previously unidentified proteins linked to biological process of interest (Aebersold & Mann, 2016). The reproducibility of proteomics between laboratories has been demonstrated by the ability to detect minor differences (Collins et al., 2017; Poulos et al., 2020). The main source of variability is the deterioration of the mass spectrometry measurement sensitivity when the instrument is approaching the need for maintenance/service (Poulos et al., 2020).

## Can exercise derived small extracellular vesicles influence ageing?

The mechanisms by which exercise promotes healthy ageing or mitigates age-related pathologies remain to be determined. It is well established that aerobic exercise promotes a transient increase in the release of sEVs into the circulation (Frühbeis et al., 2015; Vanderboom et al., 2021; Whitham et al., 2018), although the cellular origin, destination, and physiological role of these exercise EVs remain unclear. It has been proposed that sEVs act as mediators for the systematic adaptations to exercise via their ability to transport signalling molecules between tissues (Safdar & Tarnopolsky, 2018; Whitham et al., 2018). To unpack potential mechanisms by which exercise derived sEVs can influence ageing, in particular

senescence, we utilised the publicly available data sets from Whitham et al. (2018) and Vanderboom et al. (2021). It is important to note that each group of these study groups used different isolation methods to obtain sEVs, with Whitham et al. (2018) using $2 \times 20{,}000$ *g* centrifugation with results validated via $100{,}000$ *g* ultracentrifugation, whereas Vanderboom et al. (2021) combined SEC and $100{,}000$ *g* ultracentrifugation. Despite the different isolation methods employed, 102 significantly upregulated proteins following exercise were shared by both datasets. In an attempt to understand the functional effects of these proteins in the context of senescence and ageing, we cross-referenced the significantly upregulated sEV proteins following an acute bout of aerobic exercise with inhibitors of senescence identified in the CellAge Database (Avelar et al., 2020). In total, eight inhibitor proteins were identified in the exercise sEVs, with three proteins [Cu-Zn superoxide dismutase (SOD1), thioredoxin (TXN), and thymosin beta-4 (TMSB4X)] being upregulated in the sEVs from both studies (Fig. 2*A*). It is important to acknowledge that the proteins identified may play multiple or different roles from those postulated in this review; for example, GAPDH has been shown to play a role in sEV biogenesis and binds to the surface of sEVs (Dar et al., 2021).

Interestingly, TMSB4X has recently been identified as a human 'exerkine' that can potentially promote bone and neuron formation (Gonzalez-Franquesa et al., 2021). Furthermore, other studies have highlighted the potential role of TMSB4X in muscle regeneration (Spurney et al., 2010). As previously mentioned, sEVs appear to positively influence ageing, in particular senescence, via the regulation of redox homeostasis and it has previously been highlighted that sEVs can carry antioxidant enzymes: glutathione peroxidase (GPX), GST, peroxiredoxins, manganese superoxide dismutase (SOD2) or catalase (CAT) (Bodega et al., 2019). Of the identified proteins, both thioredoxin (TXN) and SOD1 are both antioxidant enzymes. Acute exercise independent of intensity has previously been shown to increase TXN expression in peripheral blood mononuclear cells (Wadley et al., 2015), whereas the suppression of TXN leads to the development of senescence (Young et al., 2010) and overexpression can extend lifespan in mice (Pérez et al., 2011). SOD1, primarily found in the cytosol, has been shown to play important roles in regulating ROS, nutrient sensing, and regeneration (Eleutherio et al., 2021; Tsang et al., 2018). SOD1 expression is increased in skeletal muscle following exercise training (Powers & Jackson, 2008), whereas SOD 1 deficient or mice exhibit accelerated ageing coupled with the accumulation of senescent cells and loss of muscle mass (Deepa et al., 2019; Zhang et al., 2017). In the context of exercise, many studies have focused on these antioxidant enzymes in

skeletal muscle. Although we cannot identify the source or sources from which these enzymes are packaged into sEVs, it is possible they are derived from skeletal muscle. One such approach from our group utilised temporal proteomics from samples obtained from catheters in the femoral vein during exercise, allowing for calculation of arterial-venous difference and subsequently net flux (Whitham et al., 2018). From this, we were able to identify 35 proteins released in sEVs from the exercising limb, of which glucose-6-phosphate dehydrogenase (G6PD) (a rate limiting enzyme in the pentose-phosphate pathway) had the largest net flux. Interestingly, an increase of ~2-fold in G6PD activity can extend healthspan in mice via an increase in NAPDH and a reduction in ROS (Nóbrega-Pereira et al., 2016). The specific tissue sources of circulating sEVs following exercise has yet to be determined. Although it may be reasonable to assume that they are derived from metabolically active tissues such as skeletal muscle, whether or not sEVs from skeletal muscle predominately have local or systemic effects remains to be determined. It has been estimated that a total of ~5% of the circulating sEV pool is derived from skeletal muscle (Estrada et al., 2022; Guescini et al., 2015). Recently, it was demonstrated using *ex vivo* tissue explants that skeletal muscle releases a greater number of sEVs than adipose tissue when normalised per unit of mass (Estrada et al., 2022). Furthermore, it was possible to demonstrate the direct release of sEVs from skeletal muscle into the circulation by means of reporter mouse combined with single EV analysis using ExoView (NanoView Biosciences, Brighton, MA, USA). Interestingly, Estrada et al. (2022) also indicated that *ex vivo* skeletal muscle sEV secretion was unchanged by contraction and potentially regulated by metabolic activity. This aligns with the data of Vanderboom et al. (2021) where single-leg resistance exercise did not promote an increase in circulating sEVs or alter sEV protein expression in contrast to aerobic exercise. Other studies have demonstrated an increase in muscle derived sEVs in the circulation following exercise as determined by increase of muscle specific microRNAs in sEVs (Guescini et al., 2015; Vechetti, Peck et al., 2021).

More studies of the functional effects of sEV skeletal muscle and other sources are required to determine whether or not they play a regulatory role in common age-related pathologies such as muscle atrophy, metabolic dysfunction, and inflammation. Indeed, it has been shown that sEVs from C2C12 myotubes are capable of regulating angiogenesis in endothelial cells (Nie et al., 2019); conversely, sEVs from senescent human primary myoblasts have been shown to impair endothelial cell function (Hettinger et al., 2021). Similar to much of the available data on sEVs from specific cell types, these *in vitro* models do not fully capitulate the complexity of the *in vivo* environment and rely on the assumption that sEVs from a given tissues or cell types interact *in vivo*.

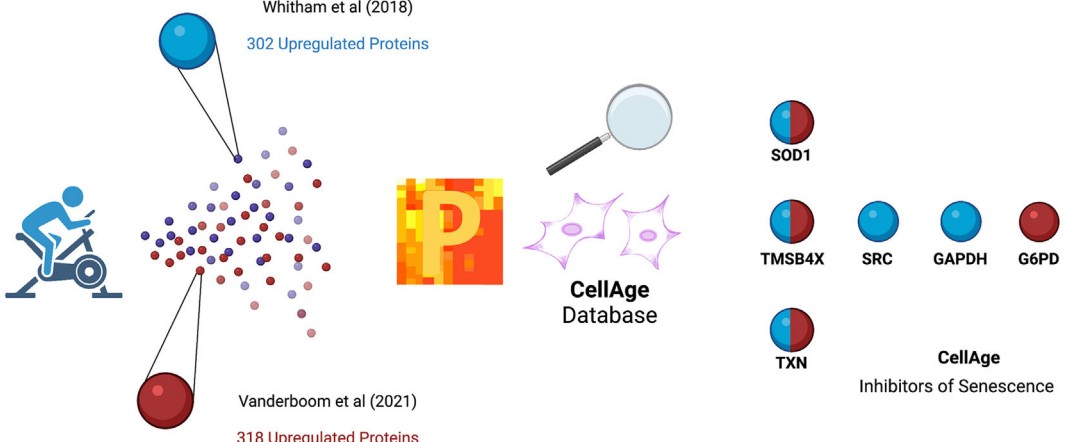

**Step 1.** Protoemic Datasets focusing on plasma sEV's in the context of exercise were downloaded from prideDB and analysed using perseus. From this we were able to identify upregulated proteins in sEV's following aerobic exercise.

**Step 2.** The identified significantly upregulated EV cargo proteins were then search against the cellAge database for inibhitors of senescence.

**Step 3.** From this we identified six protiens in sEVs upregulated by exercise, which may play a role in reducing the senscent cell load.

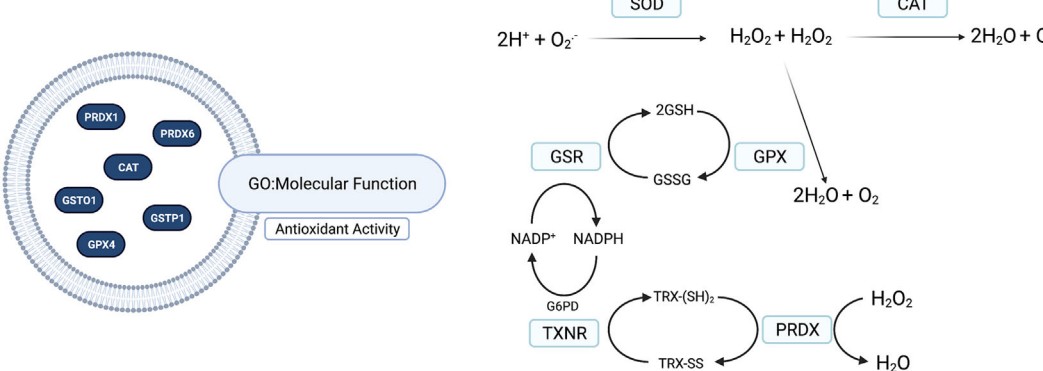

**Figure 2. Exercise sEVs contain inhibitors of cellular senescence and antioxidant enzymes**
*A*, datasets from Whitham et al. (2018) and Vanderboom et al. (2021) were analysed and cross-referenced with the CellAge Database using Perseus (Tyanova et al., 2016). The sEV proteins significantly upregulated following aerobic exercise were cross-referenced for inhibitors of senescence. Blue sEVs represent those identified by Whitham et al. (2018) and red sEVs represent those identified by Vanderboom et al. (2021). Where the sEV is both blue and red, this indicates that the protein was detected in exercise sEVs from both datasets. *B*, the significantly upregulated exercise sEV protein data were subjected to Gene Ontology molecular function analysis for antioxidant activity, identifying six antioxidant enzymes packaged within sEVs. Several proteins involved in the endogenous antioxidant defence system were identified in sEVs immediately post-exercise. This system contains several enzymes that regulate ROS to prevent molecular damage and maintain redox homeostasis. These enzymes include superoxide dismutase (SOD), catalase (CAT), glutathione peroxidase (GPX), glutathione reductase (GSR), thioredoxin oxidase (TXNR), and peroxiredoxin (PRDX). SOD and CAT remove superoxide ($O_2^{\cdot-}$) and hydrogen peroxide ($H_2O_2$), converting $O_2^{\cdot-}$ to $H_2O$, whereas GPX, GSR, TXNR, and PRDX remove $H_2O_2$ via the regulation of redox state of glutathione and thioredoxin. Created with BioRender.com.

The role of skeletal muscle derived EVs has been reviewed extensively (Darkwah et al., 2021; Rome et al., 2019; Vechetti, Valentino et al., 2021).

Significantly, of the proteins identified in comparison with the CellAge Database and Gene Ontology (http://geneontology.org) molecular function analysis for antioxidant activity, only GPX4 has a known secretory peptide as determined by SignalP-5.0 (Almagro Armenteros et al., 2019), demonstrating that extracellular vesicles present a mode by which the identified proteins enter circulation. Other proteins that act as antioxidant enzymes or play a role in redox homeostasis were also found to be abundant in exercise sEVs from both studies such as peroxiredoxin (PRDX)1, PRDX6, CAT, glutathione *S*-transferase P1, and glutathione transferase omega 1 (Fig. 2*B*). These were identified by searching the significantly upregulated post-exercise sEV proteins using Gene Ontology molecular function analysis for antioxidant activity. These findings link to well to the regenerative potential of sEVs from young cells and mice shown to reverse the senescent phenotype via antioxidant enzyme cargo proteins such as GST and peroxiredoxins (Fafián-Labora et al., 2020; Liu et al., 2019). In support of the antioxidant potential of extracellular vesicles, it was found that, after 12 weeks of high intensity interval training in individuals with pre-diabetes, there was an upregulation of antioxidant related proteins in sEVs: PRDX1, PRDX2, CAT, SOD2, and G6PD (Apostolopoulou et al., 2021). This was also accompanied by an increase in nuclear factor-erythroid factor 2-related factor 2 (NRF2) and NAD(P)H dehydrogenase [quinone] 1 (NQO1), and a decrease in NF-$\kappa$B, P38-mitogen-activated protein kinase (MAPK), and P44/42-MAPK expression in the skeletal muscle. Indeed, it has been proposed that skeletal muscle NRF2 influences the release of sEVs from skeletal muscle and the packaging of antioxidant cargo at rest and after exercise training (Gao et al., 2021). The decrease in oxidative stress often observed following exercise training can be attributed to enhanced antioxidant enzyme activity (Powers, Radak & Ji, 2016). Similarly, following acute exercise (45 min of treadmill running at 50% $\dot{V}_{O_2max}$) extracellular Cu-Zn superoxide dismutase (SOD3) and copper-transporting ATPase 1 were upregulated in human plasma sEVs (Abdelsaid et al., 2022). However, the functional effects of these antioxidant cargoes in sEVs released following exercise are yet to be determined.

## Conclusions

In this review, we have highlighted the biological relevance of sEVs in the context of cellular senescence and ageing. We have also presented a potential mechanism by which exercise can reduce the senescent cell burden and positively influence age-related pathologies. More *in vivo* studies are required to track the sEV responses to exercise across the lifespan. Given that current exercise sEV studies have focused on healthy individuals, it remains to be seen whether these positive regulators of ageing would be present in sEVs following exercise in older individuals. We have also demonstrated how publicly available proteomic datasets can be utilised to generate new hypotheses and provide a greater understanding of the physiological relevance of sEVs in a variety of different contexts. As more data are collected, the value and insight provided by these resources will grow, emphasising the importance of depositing the data to a public data repository such as ProteomeXchange (http://www.proteomexchange.org/) (Deutsch et al., 2020). It is important to note that, in this review, we have only scratched the surface of the exercise sEV proteome, and there are additional groups of sEV proteins or cargo that could contribute to healthy ageing.

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

## Additional information

### Competing interests

The authors declare that they have no competing interests.

### Author contributions

L.C.M. and M.W. were responsible for the concept of this article. Both authors approved the final version of the manuscript submitted for publication and agree to be accountable for all aspects

of the work. Both persons designated as authors qualify for authorship, and all those who qualify for authorship are listed.

## Funding

No funding was received for the present study.

## Keywords

cellular senescence, exercise, proteomics, secreted factors, small extracellular vesicles

## Supporting information

Additional supporting information can be found online in the Supporting Information section at the end of the HTML view of the article. Supporting information files available:

**Peer Review History**

