## [Peer Review History · The Journal of Physiology]

Exercise, healthy ageing, and the potential role of small extracellular vesicles

Luke Colin McIlvenna and Martin Whitham
DOI: 10.1113/JP282468

Corresponding author(s): Martin Whitham (m.whitham@bham.ac.uk)

Review Timeline:

Submission Date:	30-Nov-2021
Editorial Decision:	07-Jan-2022
Revision Received:	16-Mar-2022
Accepted:	29-Mar-2022

Senior Editor: Ian Forsythe

Reviewing Editor: Paul Greenhaff

Transaction Report:

Dear Dr Whitham,

Re: JP-TR-2021-282468 "Exercise, healthy ageing, and the potential role of small extracellular vesicles" by Luke C McIlvenna and Martin Whitham

Thank you for submitting your Topical Review to The Journal of Physiology. It has been assessed by a Reviewing Editor and by 2 expert referees and I pleased to tell you that it is considered to be acceptable for publication following satisfactory revision.

The reports are copied at the end of this email. Please address all of the points and incorporate all requested revisions, or explain in your Response to Referees why a change has not been made.

NEW POLICY: In order to improve the transparency of its peer review process The Journal of Physiology publishes online as supporting information the peer review history of all articles accepted for publication. Readers will have access to decision letters, including all Editors' comments and referee reports, for each version of the manuscript and any author responses to peer review comments. Referees can decide whether or not they wish to be named on the peer review history document.

I hope you will find the comments helpful and have no difficulty in revising your manuscript within 4 weeks.

Your revised manuscript should be submitted online using the links in Author Tasks Link Not Available. This link is to the Corresponding Author's own account, if this will cause any problems when submitting the revised version please contact us.

You should upload:

- A Word file of the complete text (including any Tables);
- An Abstract Figure, (with accompanying Legend in the article file)
- Each figure as a separate, high quality, file;
- A full Response to Referees;
- A copy of the manuscript with the changes highlighted.
- Author profile. A short biography (no more than 100 words for one author or 150 words in total for two authors) and a portrait photograph of the two leading authors on the paper. These should be uploaded, clearly labelled, with the manuscript submission. Any standard image format for the photograph is acceptable, but the resolution should be at least 300 dpi and preferably more.

- A 'Cover Art' file for consideration as the Issue's cover image;
- Appropriate Supporting Information (Video, audio or data set https://jp.msubmit.net/cgi-bin/main.plex?form_type=display_requirements#supp).

To create your 'Response to Referees' copy all the reports, including any comments from the Senior and Reviewing Editors into a Word, or similar, file and respond to each point in colour or CAPITALS. Upload this when you submit your revision.

I look forward to receiving your revised submission.

Yours sincerely,

Ian D. Forsythe
Deputy Editor-in-Chief
The Journal of Physiology
<https://jp.msubmit.net>
<http://jp.physoc.org>
The Physiological Society
Hodgkin Huxley House
30 Farringdon Lane
London, EC1R 3AW
UK
<http://www.physoc.org>
<http://journals.physoc.org>

EDITOR COMMENTS

Reviewing Editor:

This review article has been considered by two expert reviewers. Both felt the manuscript was well written and will be of interest to the readership of The Journal of Physiology, but is highly speculative. Both reviewers have raised a number of points that if addressed appropriately will improve the manuscript. In particular around the uncertainty of the methods described, not least in the context physiological adaptation. Similarly, on a related point, the proposed association between sEVs released during exercise and anti-senescence effects, which appear to be unproven at present. This speculation could be better balanced in the manuscript by disclosing to the reader the uncertainty of the methods and commenting on the nature of the association between sEVs and well described muscle ageing events, such as insulin resistance, anabolic resistance to protein nutrition, and motor unit loss. If the field is not currently sufficiently advanced to do this then this should also be disclosed. This would bring better balance. Such declaration does not need to be negative, but rather could be presented as a stimulus for future investigation.

Senior Editor:

Thank you for this interesting review. The referees and editor have some important and useful suggestions. In addition to these changes, I would be grateful if you can rethink your figures: one will be a useful abstract figure, but the others need to contain more explanatory information, so as to help your article appeal to the widest audience. In addition, please re-write the abstract to contain more factual information about the topics of your review; avoid phrases such as "This review will overview a growing field of research...", but instead make direct statements of your findings and come to a clear final concluding sentence.

REFeree COMMENTS

Referee #1:

General Comments

This is a well-written manuscript presenting a very intriguing hypothesis regarding the role of small extracellular vesicles (sEVs) that are induced by exercise and could play a role in senescence. The manuscript describes what is thought to be known in association between sEVs and senescence, the role of sEVs derived from iPSCs, and the pharmacological considerations of sEVs as a potential intervention. The text also identifies excellent resources for these data. The text puts forward a very compelling argument for their hypothesis with considerable literature support.

Regarding the manuscript, the impact of the hypothesis is extremely high. The insight provided by the review is similarly extremely high. The material presented is very timely and original. This is a review and therefore the study design question is not applicable. Lastly, the validity of the material presented is quite high and well supported by superb literature reference.

There are a few fundamental concepts associated with these considerations that would increase the balance of the presentation. First, the very definition of sEVs and therefore the proteome associated is entirely determined by the precise methods used to isolate the species. This was mentioned in the manuscript, but it is well worth more emphasis on this point. Failure to appreciate these facts has and can lead to significant misunderstandings.

Secondly, although this reviewer has not read all the proteomic references, very often, the identification of statistical significance may or may not have considered that a multitude of comparisons were made as in the high throughput proteomics. The meaning of "p" or likelihood for a type I error of 5% is modified at best by a false discovery calculation. In these contexts, the results should imply evidence for hypotheses and not simply facts. Perhaps in many of the cases, a proper experiment and null hypothesis has been constructed and the results are more credible to be facts of statistical significance. These various levels of certainty should at least be mentioned in the text to give the reader a higher level of understanding as to the level of certainty the statements of fact might represent.

Lastly, because of the prior two comments and other considerations, there is a lack of a benchmark for plasma EV proteomics methods to provide reproducibility and quantitative performance. It difficult to attribute proteomic differences to physiology or pathophysiology when reproducibility and quantitation suffer. This reviewer would agree that most of the content in the manuscript is highly likely to be factual. However, it would only be balanced if there were a clear disclaimer to

be included for the naïve reader to the field of proteomics to describe the potential for inconsistency and irreducibility, which simply reflects the state of the art of the field.

Specific Comments

1. Additional information that would describe, at a high level, the various means to obtain the sEVs by isolation methods and any implications in interpretation that might be implied for this hypothesis.

2. There are many statements of facts which associate proteins or peptides with functions. Some are the result of targeted experiments with proper null hypotheses accepted, and multiple when this condition was not present. It would be helpful to provide the reader with an indication when certainty is reduced.

3. In either case, the authors should address these basic challenges of the field for the casual reader in a single section or on a case-by-case basis within the text.

Referee #2:

The authors have provided a comprehensive but highly speculative review of the potential anti-senescence effects of small extracellular vesicles produced by exercise. They have described how sEVs may transport information and bioactive proteins from cell to cell and, via circulation, throughout the body. Senescent cells, in particular, appear to have powerful effects that may, at least partially, be mediated through sEVs. The descriptions of the parabiosis experiments, particularly recent studies provide evidence for circulating anti-senescence sEV that appear to have wide-ranging effects in most organ systems including skeletal muscle, CNS, and liver. The authors have described a series of elegant studies employing A-V differences to examine molecules that are often components of sEVs that are released from contracting skeletal muscle. Although these studies have not specifically isolated and characterized sEVs, many of the compounds identified may have anti-senescent effects. Exercise has powerful effects that counter many of the effects of aging, however, the link between sEVs released during exercise and anti-senescence is highly speculative and, as yet appear to be unproven. Clearly, aging is associated with a remarkable array of effects on skeletal muscle including growing insulin resistance, anabolic resistance to essential amino acids, loss of motor units, inflammatory mediators, periods of inactivity. As outlined in this well-written review, knowledge of sEVs biology is growing rapidly. Skeletal muscle contraction results in a large efflux of peptides and proteins into the circulation, particularly eccentric contractions - however, it is not clear the extent to which this efflux is sEVs, myokines, or uncontrolled release of a vast array of molecules. A number of neuromuscular diseases, such as muscular dystrophy, also are characterized by an exaggerated release of compounds after exercise - does this efflux from muscle contain sEVs? Might they have any positive effects? The review provides an excellent review of the somewhat limited data on muscle, exercise, and the potential anti-aging effects of sEVs. The review would be improved by commenting (or speculating) on how muscle derived sEVs may affect the already well-described effects of aging.

REQUIRED ITEMS:

Please provide (in the article file) a legend to accompany your Abstract Figure.

END OF COMMENTS

Confidential Review

30-Nov-2021

Response to Referees

JR-TR-2021-282468

Title: Exercise, healthy ageing, and the potential role of small extracellular Vesicles

Authors: Dr Luke C. McIlvenna and Dr Martin Whitham

Reviewing Editor:

This review article has been considered by two expert reviewers. Both felt the manuscript was well written and will be of interest to the readership of The Journal of Physiology, but is highly speculative. Both reviewers have raised a number of points that if addressed appropriately will improve the manuscript. In particular around the uncertainty of the methods described, not least in the context physiological adaptation. Similarly, on a related point, the proposed association between sEVs released during exercise and anti-senescence effects, which appear to be unproven at present. This speculation could be better balanced in the manuscript by disclosing to the reader the uncertainty of the methods and commenting on the nature of the association between sEVs and well described muscle ageing events, such as insulin resistance, anabolic resistance to protein nutrition, and motor unit loss. If the field is not currently sufficiently advanced to do this then this should also be disclosed. This would bring better balance. Such declaration does not need to be negative, but rather could be presented as a stimulus for future investigation.

Response: Thank you for your overview and perspective of the reviewer's comments. We have taken on board the comments and updated the manuscript accordingly.

Senior Editor:

Thank you for this interesting review. The referees and editor have some important and useful suggestions. In addition to these changes, I would be grateful if you can rethink your figures: one will be a useful abstract figure, but the others need to contain more explanatory information, so as to help your article appeal to the widest audience. In addition, please rewrite the abstract to contain more factual information about the topics of your review; avoid phrases such as "This review will overview a growing field of research...", but instead make direct statements of your findings and come to a clear final concluding sentence.

Response: Thank you for taking the time to consider our review, as requested we have created an abstract figure and redesigned the current figures to contain more explanatory information to accommodate a wider audience. In addition, we have rewritten the abstract to include more direct statements and a clear conclusion. (see lines:40-49)

Referee #1:

General Comments

Comment 1: This is a well-written manuscript presenting a very intriguing hypothesis regarding the role of small extracellular vesicles (sEVs) that are induced by exercise and could play a role in senescence. The manuscript describes what is thought to be known in association between sEVs and senescence, the role of sEVs derived from iPSCs, and the pharmacological considerations of sEVs as a potential intervention. The text also identifies excellent resources for these data. The text puts forward a very compelling argument for their hypothesis with considerable literature support.

Regarding the manuscript, the impact of the hypothesis is extremely high. The insight provided by the review is similarly extremely high. The material presented is very timely and original. This is a review and therefore the study design question is not applicable. Lastly, the validity of the material presented is quite high and well supported by superb literature reference.

Response 1: Thank you for your complimentary comments regarding the manuscript.

Comment 2: There are a few fundamental concepts associated with these considerations that would increase the balance of the presentation. First, the very definition of sEVs and therefore the proteome associated is entirely determined by the precise methods used to isolate the species. This was mentioned in the manuscript, but it is well worth more emphasis on this point. Failure to appreciate these facts has and can lead to significant misunderstandings.

Response 2: We agree that the isolation method used to obtain extracellular vesicles has the potential to influence the EV population obtained and the subsequent experimental findings. We have reemphasised the importance of choice of isolation method and have also highlighted key findings which show similarities in proteomic profile between three most commonly used isolation methods: Ultracentrifugation, size exclusion chromatography and density gradient. In addition, we have highlighted the limitation of current approaches utilising the core tetraspanin's to isolate sEVs (see lines: 291-307).

Comment 3: Secondly, although this reviewer has not read all the proteomic references, very often, the identification of statistical significance may or may not have considered that a multitude of comparisons were made as in the high throughput proteomics. The meaning of "p" or likelihood for a type I error of 5% is modified at best by a false discovery calculation. In these contexts, the results should imply evidence for hypotheses and not simply facts. Perhaps in many of the cases, a proper experiment and null hypothesis has been constructed and the results are more credible to be facts of statistical significance. These various levels of certainty should at least be mentioned in the text to give the reader a higher level of understanding as to the level of certainty the statements of fact might represent.

Response 3: Having read all the proteomic studies discussed in our review, all of them have applied a false discovery rate of <0.05 following conservative P value correction, this is now very much common practice in the field of proteomics. A key aspect of the proteomic data discussed in our review is that all the datasets are freely available online, allowing for independent re-analysis. The reproducibility in the context of EV's and exercise can be demonstrated by number of significantly upregulated proteins shared by both datasets. Between the works from Whitham and Vanderboom

a total of 102 proteins were upregulated in EVs. As for the cellular and animal work referenced in the study the changes in cargo were coupled with functional effects, although this cannot necessarily be achieved in human studies (see lines 390-396).

Comment 4: Lastly, because of the prior two comments and other considerations, there is a lack of a benchmark for plasma EV proteomics methods to provide reproducibility and quantitative performance. It difficult to attribute proteomic differences to physiology or pathophysiology when reproducibility and quantitation suffer. This reviewer would agree that most of the content in the manuscript is highly likely to be factual. However, it would only be balanced if there were a clear disclaimer to be included for the naïve reader to the field of proteomics to describe the potential for inconsistency and irreducibility, which simply reflects the state of the art of the field.

Response 4: We have now acknowledged the potential challenges of reproducibility in proteomics and highlighted the advantages of this approach (see lines: 368-379).

Specific Comments

Comment 5: 1. Additional information that would describe, at a high level, the various means to obtain the sEVs by isolation methods and any implications in interpretation that might be implied for this hypothesis.

Response 5: We have now discussed the various methods available to isolate EV's from biological samples and how this may influence experimental findings and the hypothesis we have proposed (see lines: 291-307)

Comment 6: 2. There are many statements of facts which associate proteins or peptides with functions. Some are the result of targeted experiments with proper null hypotheses accepted, and multiple when this condition was not present. It would be helpful to provide the reader with an indication when certainty is reduced.

In either case, the authors should address these basic challenges of the field for the casual reader in a single section or on a case-by-case basis within the text.

Response 6: We have added a statement clarifying that the identified proteins have multiple physiological functions (see lines: 399-402).

Referee #2:

The authors have provided a comprehensive but highly speculative review of the potential anti-senescence effects of small extracellular vesicles produced by exercise. They have described how sEVs may transport information and bioactive proteins from cell to cell and, via circulation, throughout the body. Senescent cells, in particular, appear to have powerful effects that may, at least partially, be mediated through sEVs. The descriptions of the parabiosis experiments, particularly recent studies provide evidence for circulating anti-senescence sEV that appear to have wide-ranging effects in most organ systems including skeletal muscle, CNS, and liver.

Response: Thank you for your comments.

The authors have described a series of elegant studies employing A-V differences to examine molecules that are often components of sEVs that are released from contracting skeletal muscle. Although these studies have not specifically isolated and characterized sEVs, many of the compounds identified may have anti-senescent effects. Exercise has powerful effects that counter many of the effects of aging, however, the link between sEVs released during exercise and anti-senescence is highly speculative and, as yet appear to be unproven.

Response: We have highlighted the differences in isolation methods employed by the studies from Whitham and Vanderboom, acknowledging non-EV factors could be contributing to the findings. We have acknowledged that our proposed hypothesis requires targeted studies to determine the link between sEVs released during exercise and anti-senescence effects observed following exercise. (see lines: 390-396)

Clearly, aging is associated with a remarkable array of effects on skeletal muscle including growing insulin resistance, anabolic resistance to essential amino acids, loss of motor units, inflammatory mediators, periods of inactivity. As outlined in this well-written review, knowledge of sEVs biology is growing rapidly. Skeletal muscle contraction results in a large efflux of peptides and proteins into the circulation, particularly eccentric contractions - however, it is not clear the extent to which this efflux is sEVs, myokines, or uncontrolled release of a vast array of molecules. A number of neuromuscular diseases, such as muscular dystrophy, also are characterized by an exaggerated release of compounds after exercise - does this efflux from muscle contain sEVs? Might they have any positive effects? The review provides an excellent review of the somewhat limited data on muscle, exercise, and the potential anti-aging effects of sEVs. The review would be improved by commenting (or speculating) on how muscle derived sEVs may affect the already well-described effects of aging.

Response: We only mention skeletal muscle as a potential source of EV's in one section of the review, however it is clear that EV's in circulation will be derived from multiple sources for instance from the vasculature in response to shear stress. We have added more discussion on how muscle derived EV's may or may not influence the well-described effects of ageing whilst acknowledging need for further work in this area. We also direct the readers towards other reviews which have focused specifically on the potential roles of skeletal muscle derived sEVs (see lines: 427-451).

Dear Dr Whitham,

Re: JP-TR-2022-282468R1 "Exercise, healthy ageing, and the potential role of small extracellular vesicles" by Luke Colin McIlvenna and Martin Whitham

I am pleased to tell you that your Topical Review article has been accepted for publication in The Journal of Physiology, subject to any modifications to the text that may be required by the Journal Office to conform to House rules.

NEW POLICY: In order to improve the transparency of its peer review process The Journal of Physiology publishes online as supporting information the peer review history of all articles accepted for publication. Readers will have access to decision letters, including all Editors' comments and referee reports, for each version of the manuscript and any author responses to peer review comments. Referees can decide whether or not they wish to be named on the peer review history document.

The last Word version of the paper submitted will be used by the Production Editors to prepare your proof. When this is ready you will receive an email containing a link to Wiley's Online Proofing System. The proof should be checked and corrected as quickly as possible.

All queries at proof stage should be sent to tjp@wiley.com

The accepted version of the manuscript will be published online, prior to copy editing in the Accepted Articles section.

Are you on Twitter? Once your paper is online, why not share your achievement with your followers. Please tag The Journal (@jphysiol) in any tweets and we will share your accepted paper with our 22,000+ followers!

Yours sincerely,

Ian D. Forsythe
Deputy Editor-in-Chief
The Journal of Physiology
<https://jp.msubmit.net>
<http://jp.physoc.org>
The Physiological Society
Hodgkin Huxley House
30 Farringdon Lane
London, EC1R 3AW
UK
<http://www.physoc.org>
<http://journals.physoc.org>

*** IMPORTANT NOTICE ABOUT OPEN ACCESS ***

Information about Open Access policies can be found here <https://physoc.onlinelibrary.wiley.com/hub/access-policies>

To assist authors whose funding agencies mandate public access to published research findings sooner than 12 months after publication The Journal of Physiology allows authors to pay an open access (OA) fee to have their papers made freely available immediately on publication.

You will receive an email from Wiley with details on how to register or log-in to Wiley Authors Services where you will be able to place an OnlineOpen order.

You can check if your funder or institution has a Wiley Open Access Account here <https://authorservices.wiley.com/author-resources/Journal-Authors/licensing-and-open-access/open-access/author-compliance-tool.html>

Your article will be made Open Access upon publication, or as soon as payment is received.

If you wish to put your paper on an OA website such as PMC or UKPMC or your institutional repository within 12 months of publication you must pay the open access fee, which covers the cost of publication.

OnlineOpen articles are deposited in PubMed Central (PMC) and PMC mirror sites. Authors of OnlineOpen articles are permitted to post the final, published PDF of their article on a website, institutional repository, or other free public server, immediately on publication.

Note to NIH-funded authors: The Journal of Physiology is published on PMC 12 months after publication, NIH-funded authors DO NOT NEED to pay to publish and DO NOT NEED to post their accepted papers on PMC.

EDITOR COMMENTS

Reviewing Editor:

Thank you for making the changes, which bring more perspective and balance. Congratulations on a nice review article which will be of interest and value to the readership of JP.

Senior Editor:

Thank you for an interesting Review.

REFEREE COMMENTS

Referee #2:

The authors have provided an excellent revision and comprehensive response. Although the paper remains highly speculative, the potential for sEVs is promising. This paper lays out a good rationale for how research in aging and senescence may be affected by sEV biology.

1st Confidential Review

16-Mar-2022